# Land Use and Land Cover Change Monitoring and Prediction of a UNESCO World Heritage Site: Kaziranga Eco-Sensitive Zone Using Cellular Automata-Markov Model

Nityaranjan Nath [1], Dhrubajyoti Sahariah [1], Gowhar Meraj [2], Jatan Debnath [1], Pankaj Kumar [3,*], Durlov Lahon [1], Kesar Chand [4], Majid Farooq [2], Pankaj Chandan [5], Suraj Kumar Singh [2] and Shruti Kanga [6]

1   Department of Geography, Gauhati University, Guwahati 781014, India
2   Centre for Climate Change & Water Research (C3WR), Suresh Gyan Vihar University, Jaipur 302017, India
3   Institute for Global Environmental Strategies, Hayama 240-0115, Japan
4   G.B Pant National Institute of Himalayan Environment, Himachal Regional Centre, Kullu 175126, India
5   National Development Foundation, N.D.F., Shakuntla Bhawan, Jammu 180018, India
6   Department of Geography, School of Environment and Earth Sciences, Central University of Punjab, VPO-Ghudda, Bathinda 151401, India
*   Correspondence: kumar@iges.or.jp

**Abstract:** The Kaziranga Eco-Sensitive Zone is located on the edge of the Eastern Himalayan biodiversity hotspot region. In 1985, the Kaziranga National Park (KNP) was declared a World Heritage Site by UNESCO. Nowadays, anthropogenic interference has created a significant negative impact on this national park. As a result, the area under natural habitat is gradually decreasing. The current study attempted to analyze the land use land cover (LULC) change in the Kaziranga Eco-Sensitive Zone using remote sensing data with CA-Markov models. Satellite remote sensing and the geographic information system (GIS) are widely used for monitoring, mapping, and change detection of LULC change dynamics. The changing rate was assessed using thirty years (1990–2020) of Landsat data. The study analyses the significant change in LULC, with the decrease in the waterbody, grassland and agricultural land, and the increase of sand or dry river beds, forest, and built-up areas. Between 1990 and 2020, waterbody, grassland, and agricultural land decreased by 18.4, 9.96, and 64.88%, respectively, while sand or dry river beds, forest, and built-up areas increased by 103.72, 6.96, and 89.03%, respectively. The result shows that the area covered with waterbodies, grassland, and agricultural land is mostly converted into built-up areas and sand or dry river bed areas. According to this study, by 2050, waterbodies, sand or dry river beds, and forests will decrease by 3.67, 3.91, and 7.11%, respectively; while grassland and agriculture will increase by up to 16.67% and 0.37%, respectively. The built-up areas are expected to slightly decrease during this period (up to 2.4%). The outcome of this study is expected to be useful for the long-term management of the Kaziranga Eco-Sensitive Zone.

**Keywords:** KNP; eco-sensitive zone; CA-Markov; geospatial; LULC change dynamics





## 1. Introduction

The land use change of the Earth's surface brought on by both natural and human activity is known as land use and land cover (LULC) change [1–3]. With the advancement of geospatial models, monitoring the status and dynamics of LULC using open-access remote sensing data has emerged as an effective tool for conservation and management [4–7]. Cellular Automata-Markov (CA-Markov) models are widely used to predict LULC since they combine the principles of cellular automata (CA) with Markov chains [8,9], and their rigorous structure is particularly effective in considering the natural complexity of the landscape [10,11].

In this study, we explored past and future LULC changes in the Kaziranga Eco-Sensitive Zone, an area that includes the Kaziranga National Park and a 10 km buffer area

using CA-Markov models. The prediction of LULC has been carried out for the years 2030, 2040, and 2050. We used supervised classification for assessing the past changes in the LULC in the Kaziranga National Park and the CA-Markov model for assessing future changes. The region mainly faces many challenges regarding the growing human intervention in this eco-sensitive zone, which covers a largely populated area. This area plays a vital role in maintaining the ecosystem [12]. Because of constant human influence, this area is slowly losing its natural characteristics. In particular, because of growing tourism activity, the southern part of the region now sees an increase in the surface that is occupied by built-up areas and roads, which represent a serious disturbance to the free movements of the animals [13]. We hope that the findings of this study will help the relevant authorities and policymakers to practice more environmentally responsible land and natural hazard management.

The study area is one of the famous national parks in India, and biologically, it has a lot of significance. Kaziranga is regarded as one of the oldest wildlife refuges in the world. The park's contribution in saving the Indian one-horned rhinoceros from the brink of extinction at the turn of the 20th century to harboring the single largest population of this species is a spectacular conservation achievement. Therefore, LULC change in this national park is a burning issue concerning conservation.

The Brahmaputra River passes through this region, making it prone to devastating floods on an annual basis [14]. It is believed that the findings of this study shall help the relevant authorities and policymakers to practice more environmentally responsible land and natural hazard management. The study will aid in policymaking and constant monitoring of land changes over a short time frame by tracking and analyzing the region's current and future land use and land cover scenarios.

## 2. Materials and Methods

### 2.1. Study Area

The Kaziranga Eco-Sensitive Zone lies at the intersection of 26.27 N to 26.51 N and the longitudinal extent of 92.54 E to 93.42 E in the Golaghat, Karbi Anglong, Nagaon, and Biswanath districts of Assam (Figure 1). The study area consists of the KNP and the eco-sensitive zone within a 10km radius around it. The Jia Bharali River bounds it towards the west, National Highway 15 towards the north, Karbi Anglong hills towards the south, and the Dhansiri River towards the east. The average length of the study area is 70 km, and the average width is 31 km, with an area of 2214 km$^2$. Physiographically, the study area consists mainly of alluvial flood plains of the Brahmaputra River and the Karbi Anglong range hilly areas. Apart from this, riverine lakes (locally known as Beels) and exposed sand or dry river bed bars (locally known as Chapories) also constitute a significant physiographic area. The Kaziranga–Karbi Anglong area falls in the sub-tropical monsoon climate. The temperature in the area ranges from 7 to 38 degrees Celsius. The annual average rainfall is 250 cm. The drainage system of the study area is controlled by the Brahmaputra River and its various tributaries. The Brahmaputra River flows through the middle part of the study area and carries large amounts of sediment, resulting in the formation of sand or dry river beds bar within the channel. The study area comprises various riverine lakes (Beels) and water ecosystems. Every year, the Brahmaputra River and its tributaries flood the core part of the KNP and surrounding areas. However, one of the advantages of the drainage system and the annual floods is that it helps maintain the area's natural water ecosystem. Moreover, the monsoonal climate, flat floodplains, and fertile soil of the area have provided favorable conditions for the rich growth of vegetation.

There are four main types of vegetation in the study area: tropical mixed deciduous forests, tropical semi-evergreen forests, alluvial savanna woodlands, and alluvial inundated grasslands. The grasslands make up the majority of the central portion of the study area. At the same time, the tall grasses and reeds make up the majority of the vegetation in this region (also known as elephant grass). Shorter grasses predominate in the Beels and

low-lying areas. There are tropical mixed deciduous and semi-evergreen forests in the southern part of the study area, which is located in the Karbi Anglong hills.

**Figure 1.** Location map of the Kaziranga Eco-Sensitive Zone.

## 2.2. Datasets

Landsat TM 1990, TM 2000, ETM+ 2010, and Landsat OLI 2020 were used (Table 1). The freely accessible Landsat imageries were downloaded from the United States Geological Survey (USGS) website (https://earthexplorer.usgs.gov/ accessed on 31 Dec 2021). The imageries were exclusively collected for the post-monsoon seasons to obtain less cloud cover data. Further, the temperature and rainfall data were downloaded from the NASA POWER website (https://power.larc.nasa.gov/data-access-viewer/ accessed on 31 December 2021) to assess the impact of climate change on land use and land cover.

**Table 1.** Details of the satellite images used in the present study.

| S. No | Sensor | Path/Row | Acquisition Date | Spatial Resolution |
|-------|--------|----------|------------------|--------------------|
| 1 | TM | 136/41 | 30-12-1990 | 30 |
| 2 | TM | 136/41 | 08-01-2000 | 30 |
| 3 | TM | 136/41 | 15-01-2010 | 30 |
| 4 | OLI | 136/41 | 04-02-2020 | 30 |

### 2.2.1. Image Preprocessing

The image preprocessing stage included bad line detection and restoration techniques, geometric rectification or image registration, topographic correction, atmospheric correction, and radiometric correction to attain better classification and accuracy of the images using ArcGIS version 10.8 [15,16]. The acquired Landsat datasets were projected using the Universal Transverse Mercator (UTM) projection with zone 47 North and the World Geodetic System (WGS84) 84 datum. Additionally, all the satellite images were composed by the RGB color scheme, making it easy to view and distinguish every object on the Earth's surface in detail [16].

### 2.2.2. Image Classification

In the present study, six LULC classes were considered: waterbody, sand or dry river beds, forest, grassland, agricultural land, and built-up areas, to identify the natural habitat and human interference in the KNP. The image classification process was carried out using a supervised classification approach, and the imagery was classified by the maximum likelihood algorithm (MLC), producing pixel-by-pixel LULC maps of the study area [17–20].

### 2.3. LULC Change Detection

Temporal changes in the LULC were monitored and analyzed with a post-classification change detection method by employing a change matrix (raster polygon). The change matrix depicts the LULC change in each period starting from 1990 to 2020. LULC changes for the years 2030, 2040, and 2050 were also predicted [21]. The following equation was used to calculate the degree of change for each class:

$$C_i = L_i - B_i \tag{1}$$

where $C_i$ denotes the magnitude of change in class $i$, $B_i$ denotes the base image, and $L_i$ represents the latest image.

We calculated the percentage of change for each LULC category using the following formula:

$$P_i = \frac{L_i - B_i}{B_i} \times 100 \tag{2}$$

where $P_i$ denotes the percentage of change in class $i$, $B_i$ denotes the base image and $L_i$ represents the latest image.

### Accuracy Assessment

Accuracy and model validation are essential for a reliable LULC study. In the context of classified maps, accuracy is typically defined as the extent to which classifications are accurate [22]. In the present study, the accuracy of the classified maps was assessed using kappa and overall accuracy statistics. In this process, satellite maps, Google Earth imagery, and GPS points were used to support the validation process. The confusion matrix was generated after combining the ground reference data with the classified LULC map for classification accuracy. The producer accuracy, user accuracy, overall accuracy, and kappa accuracy were all calculated. The overall accuracy is calculated as the total number of correctly classified pixels divided by the total number of pixels in the error matrix; whereas the Kappa coefficient defines the degree of agreement between the classified map and the reference data [22,23]. These two measures were computed using the following formulas:

$$\text{Overall accuracy}: \frac{\sum_{i=1}^{r} x_{ii}}{N} * 100 \tag{3}$$

where $r$ is the number of rows in the matrix, $x_{ii}$ expresses the total number of correctly

classified pixels in row *i* and column *i*, and *N* is the total number of pixels in the matrix table.

$$\text{Kappa accuracy}: \frac{N\sum_{i=1}^{r} x_{ii} - \sum_{i=1}^{r}(x_{i+} * x_{+i})}{N^2 - \sum_{i=1}^{r}(x_i + * x +_i)} \tag{4}$$

where *r* is the number of rows in the matrix, $x_{ii}$ expresses the total number of correctly classified pixels in row *i* and column *i*, $x_{i+}$ and $x_{+i}$ are the marginal totals of row *i* and column *i*.

### 2.4. Prediction of LULC Change Using the CA-Markov Chain Model

We used CA-Markov models to simulate the most likely future scenarios of LULC changes [23,24]. IDRISI Terrset software was applied to the transition suitability image and LULC maps from 1990, 2000, 2010, and 2020 [25,26]. Subsequently, the transition probability map was generated from 2000 to 2010 to simulate the LULC map for 2020. The period 2010–2020 was used to simulate the LULC map of 2030, 2040, and 2050 using the Markovian Transition Estimator technique. The multi-criteria evaluation (MCE) module's constraints and variables were used to calculate the transition suitability image [27,28]. Finally, the base map, the transition suitability map, and the transition probabilities map were used to model the LULC for the years 2030, 2040, and 2050 using the Cellular Automata/Markov Change Prediction technique.

The Cellular Automata (CA) model equation is shown in the following Equation (1) [2,29].

$$S(t, t+1) = f(S(t), N) \tag{5}$$

where *S* (*t* + 1) is the system status at the time of (*t*, *t* + 1), and function is the state probability of any time (*N*)

This model is frequently used for LULC monitoring, ecological modeling, change simulation, and estimating future land use change and stability in a given region. Expected shifts in LULC for the future are calculated as:

$$S(t, t+1) = P_{ij} \times S(t) \tag{6}$$

where *S* (*t*) is the system status at time *t*, *S* (*t* + 1) is the system status of time *t* +, and *Pij* is the transition probability matrix in a state which is computed as follows [20,30]:

$$\|P_{ij}\| = \left\| \begin{matrix} P1,1 & P1,2 & P1,N \\ P1,1 & P1,2 & P2,N \\ \cdots\cdots\cdots \\ PN,1 & PN,2 & PN,N \end{matrix} \right\| \tag{7}$$

$$(0 \le P_{ij} \le 1)$$

where *P* is the transition probability, $P_{ij}$ stands for the probability of transforming from present state *i* to another state *j* in succeeding time, and *PN* is the state probability of any time. A low transition will have a probability close to 0, while a high transition has a probability close to 1 [20]. Markov Chain estimates how much land area would change from the previous year to the projected year.

Calibration and Validation of the Model

After comparing the real LULC (2020) with the simulated LULC (2020), chi-square ($x^2$) test statistics were created for the accuracy assessment of the projected LULC images. However, this is not necessarily supported by the agreement regarding the spatial distribution of the LULC classes at the study site. A more sophisticated method, called agreement and disagreement statistics, was applied between the two images to address this problem [28–32]. To analyze the model, it is necessary to grasp the Disagreement Quantity and Disagreement Grid Cell elements. This type of validation technique provides

an assessment of the degree of agreement or discrepancy between the projected and actual LULC maps. Quantity (changes or persistence) and allocation are the key areas where the two categories' maps diverge. Disagreement by quantity is the difference between two images caused by an improper combination of the total LULC category proportions. The difference between the two images is known as the allocation disagreement, and it results from the inappropriate integration of the spatial allocations of all land cover map categories. The overall methodology is shown in Figure 2.

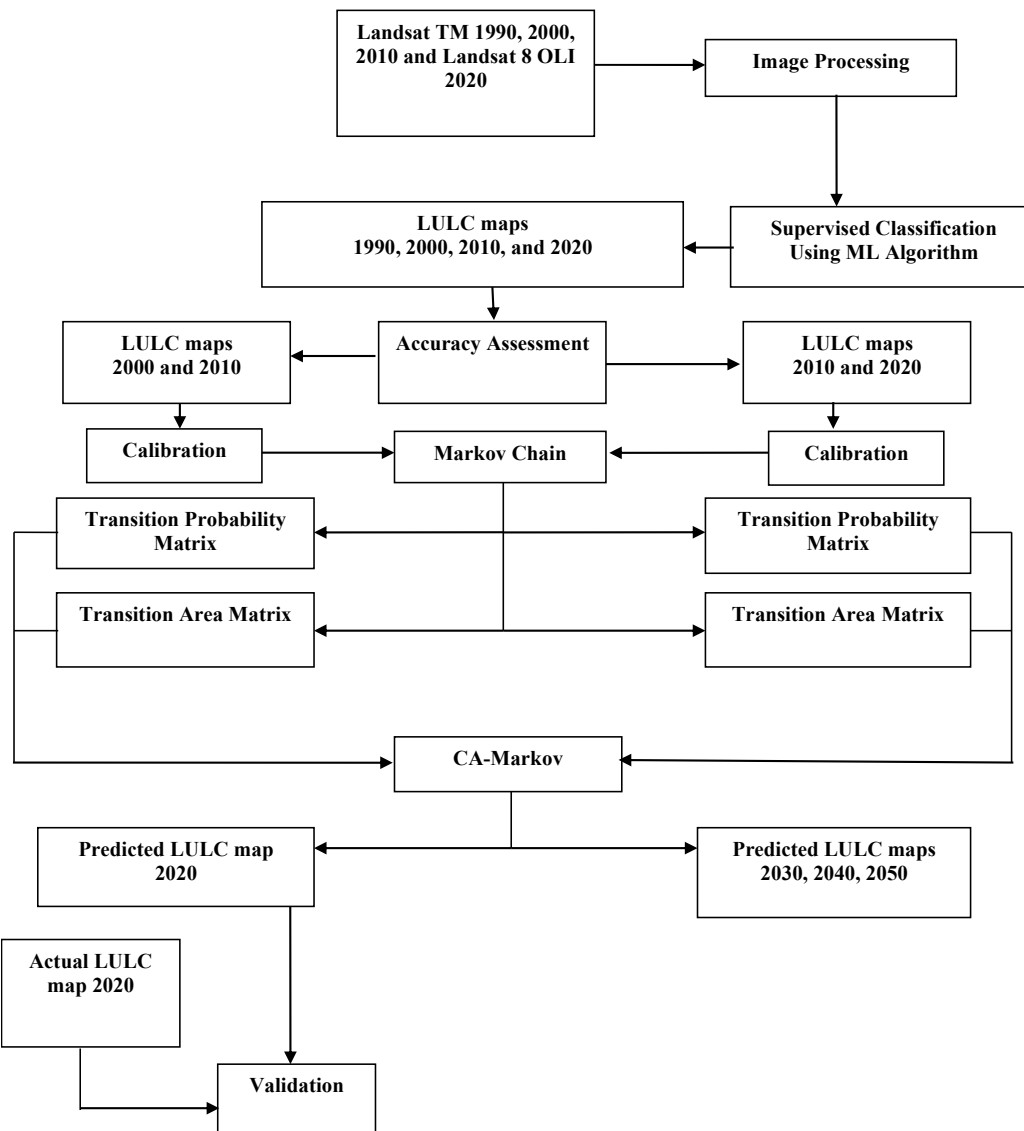

**Figure 2.** Flowchart of the methodology applied in the present study.

## 3. Results and Discussion

### 3.1. Assessing the Spatio-Temporal Distribution of LULC Change

In order to monitor and analyze temporal and spatial shifts in LULC, this study divided the landscape into six distinct types: waterbodies, sand or dry river beds, grassland, forest, built-up area, and agricultural land (Figures 3 and 4). During the study period, the eco-sensitive zone surrounding Kaziranga was dominated by forests and grasslands. Beginning in 1990, the forest cover began to shrink, but in 2020, it increased (Table 2).

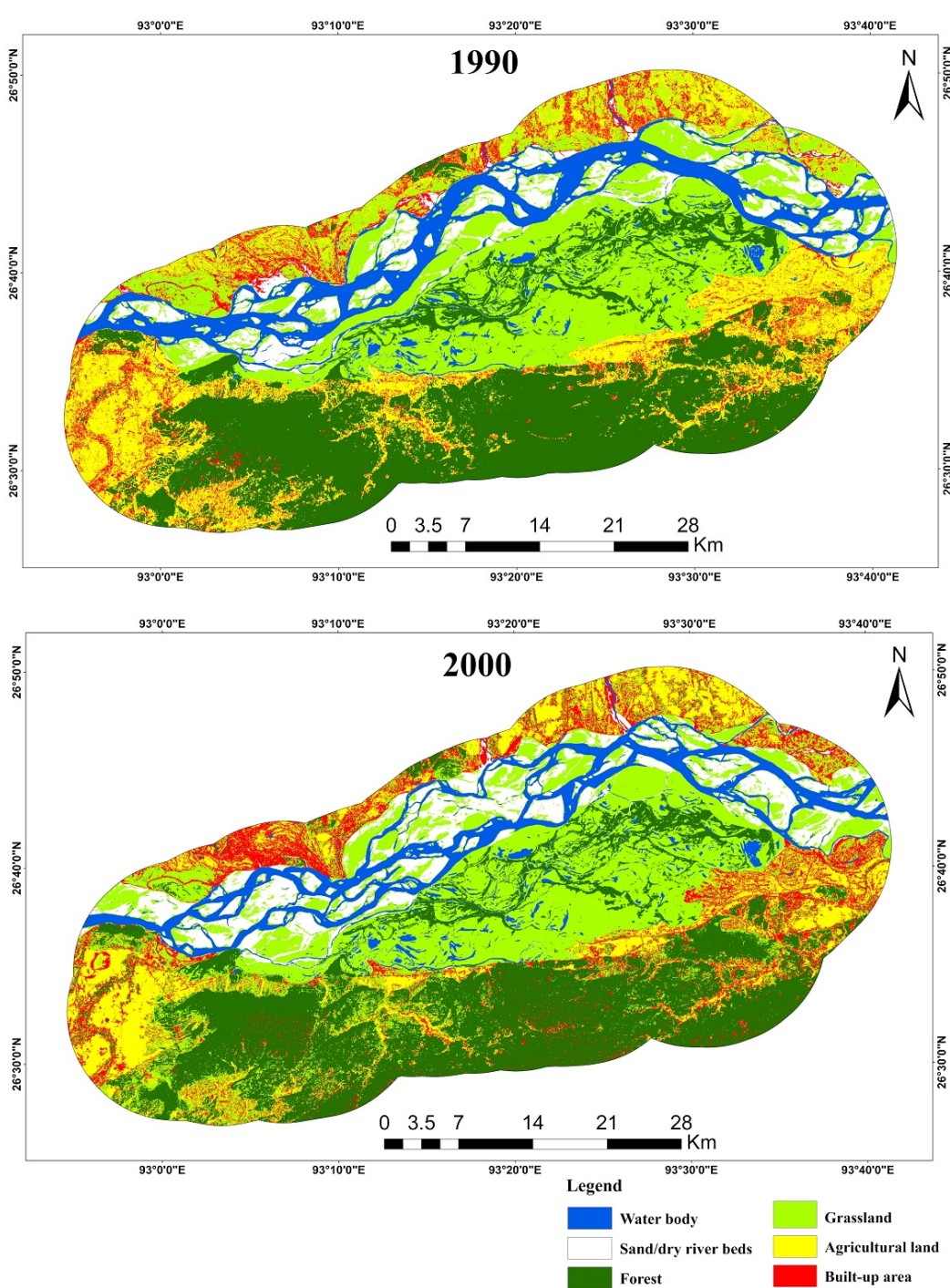

**Figure 3.** LULC change distribution from 1990 to 2000 at Kaziranga Eco-Sensitive Zone.

Grassland expansion occurred concurrently with forest expansion, particularly between 1990 and 2010. Grassland covered a larger percentage of the landscape between 1990 and 2010, but by 2020, that percentage had dropped. The number of waterbodies in the study area declined dramatically between 1990 and 2010, with a modest recovery between 2010 and 2020. More sand is being deposited by the Brahmaputra River, especially near its northern border with Kaziranga National Park, as the sand or dry river bed cover has increased continuously from 6.62% in 1990 to 13.48% in 2020 [14]. In total, 12.15% of farmland disappeared during the study period. The built-up area is the most important category in this analysis because it has almost doubled from 1990 to 2020.

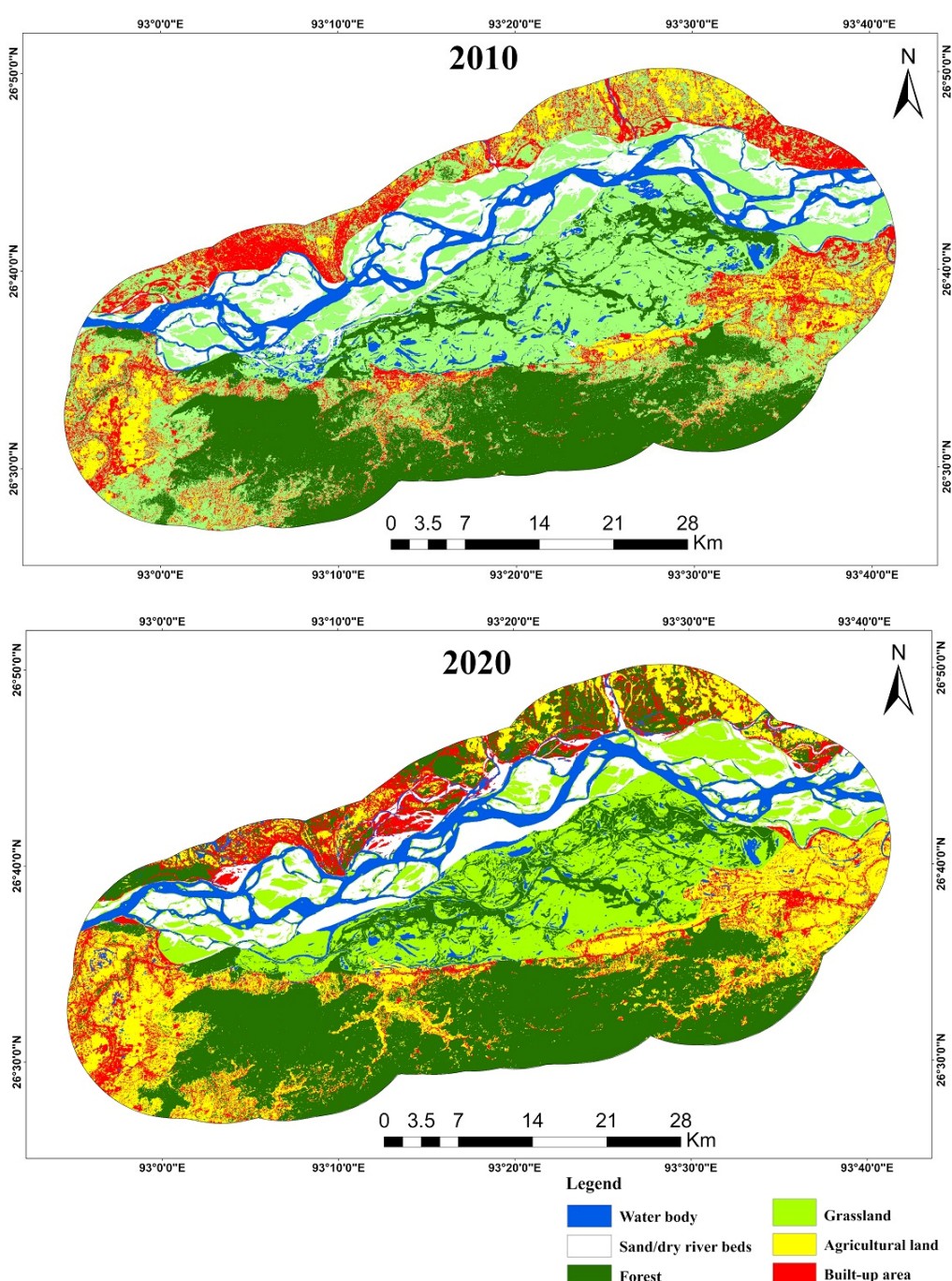

**Figure 4.** LULC distribution change from 2010 to 2020 at Kaziranga Eco-Sensitive Zone.

Many LULC shifts occurred in the study area during the 1990s, the 2000s, and the 2010s (Table 3). There was an increase in the area with sand and dry river beds, grassland, and built-up land between 1990 and 2010. The area covered with waterbodies, forests, and agricultural land was reduced. In the period 2000 to 2010, the area occupied by sand or dry river beds, grassland, and built-up areas expanded, whereas areas occupied by waterbodies, forest, and agriculture shrank. Between 2010 and 2020, the areas with waterbodies, sand or dry river beds, forests, and built-up surfaces increased, while the areas covered with grassland and agricultural land decreased. Between 1990 and 2020, the amount of land covered with built-up surfaces, forest, and sand or dry river beds increased, while the areas covered with waterbodies, grassland, and agricultural land shrank.

**Table 2.** Area of the different classified LULC classes of the study area.

| LULC Class | 1990 | | 2000 | | 2010 | | 2020 | |
|---|---|---|---|---|---|---|---|---|
| | Area (km²) | Area (%) | Area (km²) | Area (%) | Area (km²) | Area (%) | Area (km²) | Area (%) |
| Waterbody | 229.77 | 10.55 | 199.74 | 9.17 | 164.29 | 7.54 | 187.5 | 8.61 |
| Sand/Dry river Beds | 144.15 | 6.62 | 203.07 | 9.32 | 205.41 | 9.43 | 293.66 | 13.48 |
| Forest | 655.69 | 30.12 | 625.87 | 28.73 | 557.41 | 25.59 | 701.34 | 32.19 |
| Grassland | 551.17 | 25.32 | 685.92 | 31.48 | 803.93 | 36.9 | 496.26 | 22.78 |
| Agricultural Land | 407.51 | 18.72 | 202.29 | 9.29 | 145.07 | 6.66 | 143.12 | 6.57 |
| Built-up Area | 188.84 | 8.67 | 261.68 | 12.01 | 302.4 | 13.88 | 356.97 | 16.38 |

**Table 3.** Period-wise transformations of LULC class in the Kaziranga Eco-Sensitive Zone.

| LULC Class | 1990–2000 | | 2000–2010 | | 2010–2020 | | 1990–2020 | |
|---|---|---|---|---|---|---|---|---|
| | Area (km²) | Area (%) | Area (km²) | Area (%) | Area (km²) | Area (%) | Area (km²) | Area (%) |
| Waterbody | −30.03 | −13.07 | −35.45 | −17.75 | 23.21 | 14.13 | −42.27 | −18.4 |
| Sand/Dry river beds | 58.92 | 40.87 | 2.34 | 1.15 | 88.25 | 42.96 | 149.51 | 103.72 |
| Forest | −29.82 | −4.55 | −68.46 | −10.94 | 143.93 | 25.82 | 45.65 | 6.96 |
| Grassland | 134.75 | 24.45 | 118.01 | 17.2 | −307.67 | −38.27 | −54.91 | −9.96 |
| Agricultural Land | −205.22 | −50.36 | −57.22 | −28.29 | −1.95 | −1.34 | −264.39 | −64.88 |
| Built-up Area | 72.84 | 38.57 | 40.72 | 15.56 | 54.57 | 18.05 | 168.13 | 89.03 |

The transformation of LULC class from one class to another class for the study period is shown in Table 4 and Figure 4. From 1990 to 2000, the transformation of agricultural land into built-up areas was among the highest, while the lowest LULC conversion was observed from sand or dry river beds to forest land in the Kaziranga Eco-Sensitive Zone. From 2000–2010 the conversion of forest area to grassland was among the highest. On the contrary, the transformation of the area covered with waterbodies to agricultural land was the lowest. From 2010 to 2020, the transformation of the grassland area to forest was the highest, and the agricultural land to grassland was the lowest among all the classes. The overall study period of 1990 to 2020 shows that the agricultural land to built-up area conversion rate was among the highest, and the forest to sand or dry river beds conversion rate was among the lowest (Figures 5 and 6).

**Table 4.** Period-wise transformations of LULC class of the Kaziranga eco-sensitive zone.

| LULC Class Transformation | 1990–2000 | 2000–2010 | 2010–2020 | 1990–2020 |
|---|---|---|---|---|
| | Area (km²) | Area (km²) | Area (km²) | Area (km²) |
| Waterbody—Sand/Dry river beds | 76.73 | 67.5 | 47.16 | 68.08 |
| Waterbody—Forest | 0.86 | 6.28 | 1.55 | 10.46 |
| Waterbody—Grassland | 60.69 | 60.88 | 48.01 | 57.2 |
| Waterbody—Agricultural Land | 0.25 | 0.14 | 0.45 | 4.08 |
| Waterbody—Built-up Area | 2.86 | 6.76 | 5.46 | 12.7 |
| Sand/Dry river beds—Waterbody | 37.18 | 41.93 | 45.71 | 30.36 |
| Sand/Dry river beds—Forest | 0.1 | 0.33 | 4.52 | 5.17 |
| Sand/Dry river beds—Grassland | 49.47 | 76.91 | 55.68 | 43.12 |
| Sand/Dry river beds—Agricultural Land | 0.47 | 0.33 | 0.79 | 3.03 |
| Sand/Dry river beds—Built-up Area | 2.83 | 11.34 | 15.11 | 10.25 |
| Forest—Waterbody | 1.91 | 0.83 | 8.29 | 4.42 |
| Forest—Sand/Dry river beds | 0.74 | 0.47 | 0.24 | 1.71 |
| Forest—Grassland | 65.67 | 121.49 | 18.08 | 18.51 |

**Table 4.** *Cont.*

| LULC Class Transformation | 1990–2000 | 2000–2010 | 2010–2020 | 1990–2020 |
|---|---|---|---|---|
| | Area (km²) | Area (km²) | Area (km²) | Area (km²) |
| Forest—Agricultural Land | 2.03 | 0.38 | 13.15 | 57.68 |
| Forest—Built-up Area | 25.16 | 20.41 | 7.12 | 15.77 |
| Grassland—Waterbody | 63.64 | 59.14 | 58.23 | 60.32 |
| Grassland—Sand/Dry river beds | 62.28 | 58.19 | 55.04 | 55.68 |
| Grassland—Forest | 12.9 | 46.49 | 173.28 | 77.43 |
| Grassland—Agricultural Land | 24.89 | 7.43 | 143.81 | 28.06 |
| Grassland—Built-up Area | 31.3 | 76.06 | 83.71 | 34.89 |
| Agricultural Land—Waterbody | 2.8 | 0.48 | 1.63 | 12.48 |
| Agricultural Land—Sand/Dry river beds | 2.05 | 0.62 | 0.12 | 3.47 |
| Agricultural Land—Forest | 32.03 | 0.39 | 5.2 | 54.58 |
| Agricultural Land—Grassland | 100.5 | 26.32 | 0.06 | 4.00 |
| Agricultural Land—Built-up Area | 116.11 | 55.32 | 15.8 | 96.45 |
| Built-up Area—Waterbody | 1.46 | 1.95 | 2.04 | 2.16 |
| Built-up Area—Sand/Dry river beds | 0.13 | 0.56 | 0.04 | 0.08 |
| Built-up Area—Forest | 0.18 | 0.84 | 0.45 | 0.97 |
| Built-up Area—Grassland | 0.43 | 0.59 | 0.07 | 0.82 |
| Built-up Area—Agricultural Land | 0.54 | 0.08 | 0.3 | 0.92 |

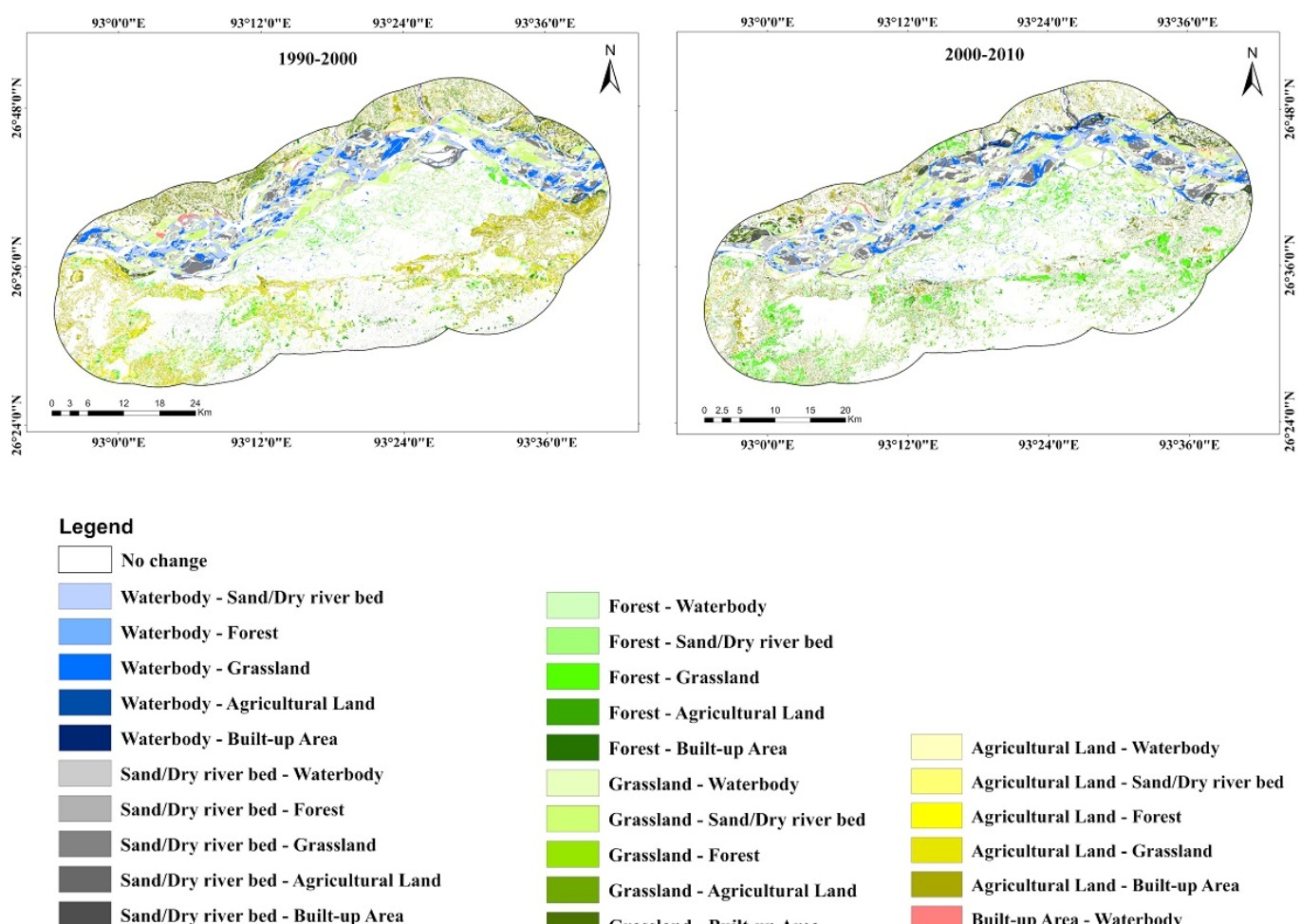

**Figure 5.** Change map of different LULC classes between 1990–2000 and 2000–2010.

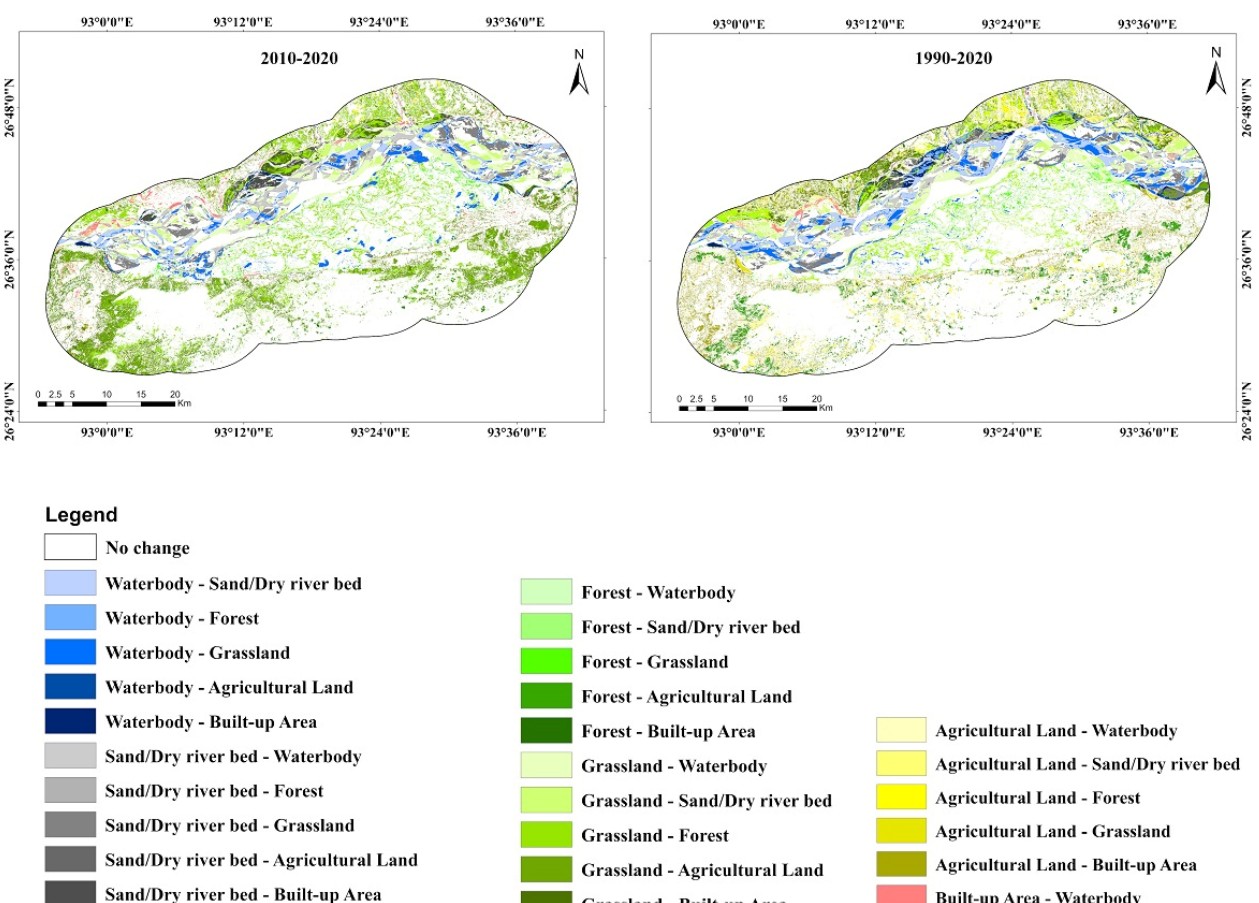

**Figure 6.** Change map of different LULC classes between 2000–2010 and 2010–2020.

### 3.2. Accuracy Assessment

In all LULC categorized maps, both kappa accuracy and overall accuracy retained an accuracy of 85% and above, which is reliable for further analysis (Table 5) [20,31].

**Table 5.** Accuracy assessment of the classified LULC images.

| LULC Classes | 1990 | | 2000 | | 2010 | | 2020 | |
|---|---|---|---|---|---|---|---|---|
| | Producers Accuracy | Users Accuracy | Producers Accuracy | Users Accuracy | Producers Accuracy | Users Accuracy | Producers Accuracy | Users Accuracy |
| Waterbody | 94.00 | 97.00 | 71.74 | 74.16 | 99.00 | 100.00 | 98.15 | 99.00 |
| Sand/Dry river beds | 78.00 | 87.00 | 83.02 | 74.58 | 94.00 | 82.00 | 90.12 | 88.25 |
| Forest | 94.00 | 81.00 | 96.88 | 100.00 | 100.00 | 83.00 | 93.23 | 89.56 |
| Grassland | 95.00 | 94.00 | 95.00 | 100.00 | 96.00 | 100.00 | 91.36 | 86.63 |
| Agricultural Land | 97.00 | 86.00 | 87.89 | 74.36 | 97.00 | 91.00 | 94.69 | 87.78 |
| Built-up Area | 85.00 | 80.00 | 75.00 | 72.00 | 89.00 | 85.00 | 87.00 | 86.00 |
| *Overall Accuracy* | 91.00% | | 86.00% | | 94.00% | | 92.00% | |
| *Kappa Accuracy* | 90.00% | | 85.00% | | 93.00% | | 91.00% | |

### 3.3. Future LULC Change Projections

#### 3.3.1. LULC Transition Probabilities

The transition probability matrix for the 2020 prediction of the LULC map was created using the classified maps of 2000 and 2010. Similarly, utilizing classified maps from 2010 and 2020, the probability matrix for forecasting the LULC map for 2030, 2040, and 2050 was also created. The simulated map is shown in Figures 7 and 8. The classified map of 2020 was used to validate the anticipated map for that year (reference imagery). The 6 × 6 LULC

class matrix table was used to examine the changes. In this matrix table, earlier LULC classes during the time (t1) were shown in the rows, while the later LULC classes during the time (t2) were shown in the columns. These matrices show the prior probabilities in the LULC classes, which were subsequently applied to predict the LULC classes based on pertinent years.

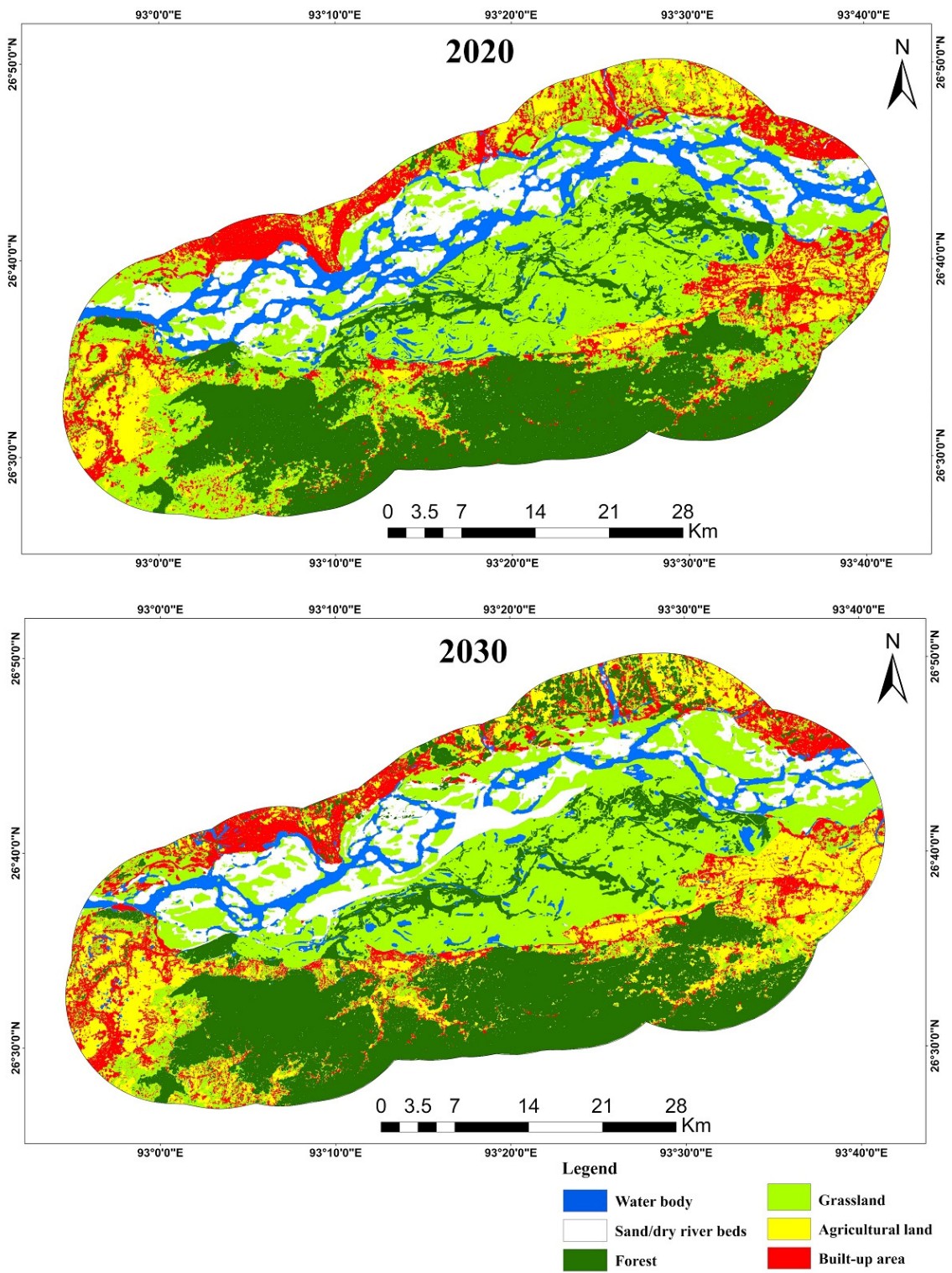

**Figure 7.** Predicted LULC of Kaziranga Eco-Sensitive Zone for the years 2020 and 2030.

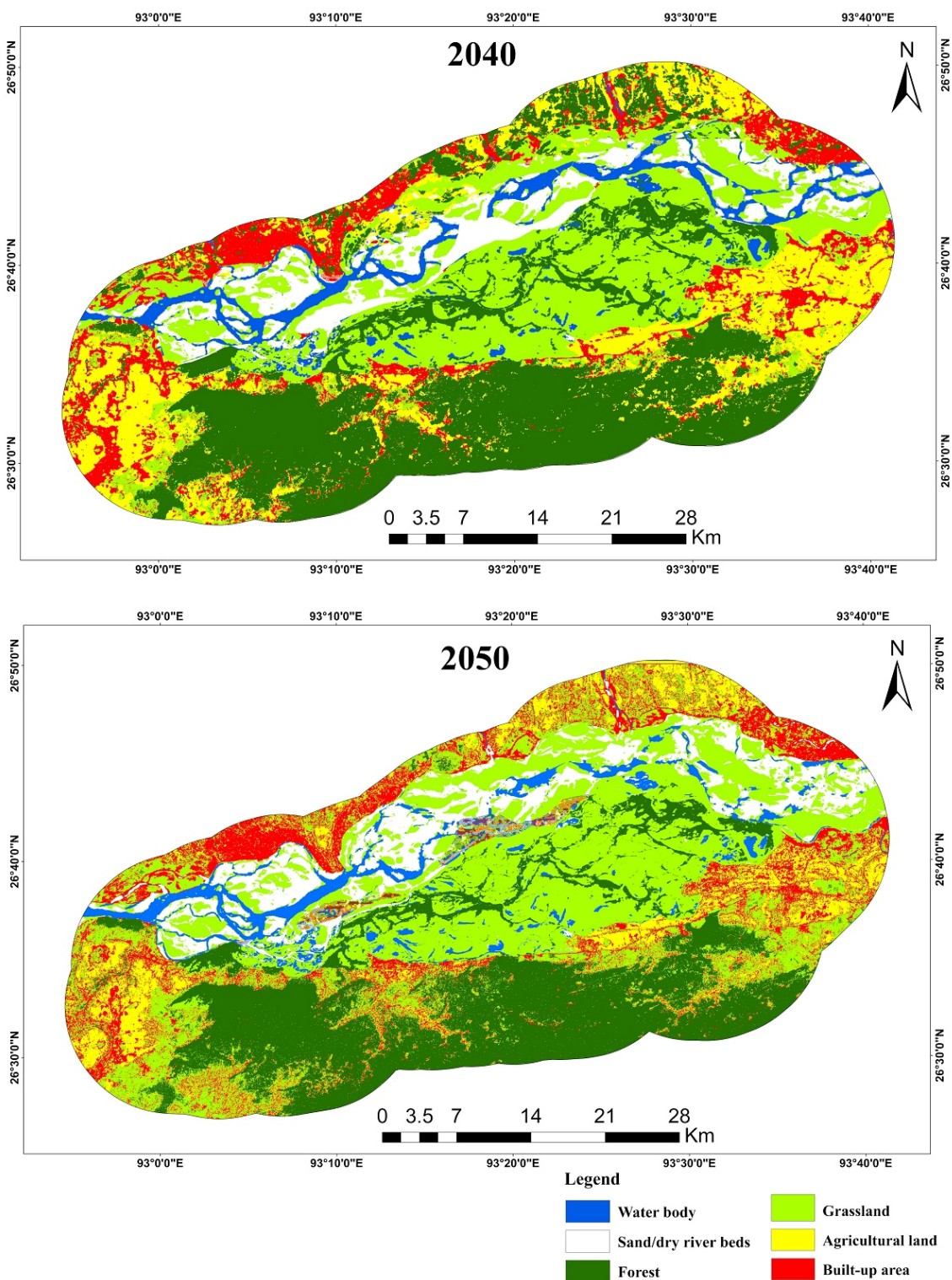

**Figure 8.** Predicted LULC of Kaziranga Eco-Sensitive Zone for the years 2040 and 2050.

This study forecasted 30 years of future prediction based on the existing data sources. Table 6 depicts the transformation in LULC categories from 2020 to 2030. The analysis indicates that there is a 34.9% chance that the area classified as a waterbody will remain that way and that there is a 29.79% chance that it will be converted to sand, 1.11% to forest, 30.44% to grassland, 0.03% to agricultural land, and 3.46% to a built-up area. The chances of the grassland changing into a waterbody are 7.45%, while the chances of it staying the same are 34.58%. Areas buried by sand or dry river beds have a 37.07%

chance of remaining dry, while these areas have a 23.51% chance of becoming a body of water. Furthermore, there is a 91.38% chance that the forested area will not change at all between 2020 and 2030, and a 3.29% chance that it will become grassland. The likelihood of agricultural land remaining unchanged during this time period is 83.8%, among the lowest of all classes, while the likelihood of agricultural land becoming a built-up area is 11.25%. The chances of no change in the urban footprint during the upcoming years are 3.8%.

**Table 6.** Probability of LULC changes in 2030 by percentage.

| 2020/2030 | Waterbody | Sand/Dry River Beds | Forest | Grassland | Agricultural Land | Built-Up Area |
|---|---|---|---|---|---|---|
| Waterbody | 34.9 | 29.79 | 1.11 | 30.44 | 00.3 | 3.46 |
| Sand/Dry river beds | 23.51 | 37.07 | 2.36 | 28.89 | 00.42 | 7.75 |
| Forest | 1.01 | 0.04 | 91.38 | 3.29 | 02.5 | 1.29 |
| Grassland | 7.45 | 7.04 | 21.99 | 34.58 | 18.02 | 10.91 |
| Agricultural Land | 1.16 | 1.00 | 3.66 | 00.03 | 83.8 | 11.25 |
| Built-up Area | 0.20 | 0.04 | 0.1 | 00.02 | 00.00 | 38.4 |

The result also shows that 454,751 pixels from other classified groups are predicted to become agricultural land. Furthermore, it implies that agricultural land would mostly be converted to the class of built-up areas (47,346 pixels). The main change in the grassland class is the transformation of grassland into a built-up area of 10,642 pixels. However, a minor change is projected for this class to become a forest, sand or dry river beds, and a waterbody. The maximum area occupied by being built-up (104,727 pixels) would remain unchanged. The waterbody class is expected to remain unchanged (72,855 pixels), whereas 62,192, 2320, 63,547, 636, and 7213 pixels would be converted into the sand or dry river bed, forest, grassland, agricultural land, and built-up areas, respectively. The area under the sand or dry river beds class would also transition, with 48,195 pixels expected to become a waterbody and roughly 76,006 pixels expected to remain predominantly unchanged. Moreover, 72,855 pixels of the waterbody class are expected to remain unchanged (Table 7).

**Table 7.** Expected transition in 2030 by number of pixels.

| 2020/2030 | Waterbody | Sand/Dry River Beds | Grassland | Forest | Agricultural Land | Built-Up Area |
|---|---|---|---|---|---|---|
| Waterbody | 72,855 | 62,192 | 2320 | 63,547 | 636 | 7213 |
| Sand/Dry river beds | 48,195 | 76,006 | 4839 | 59,230 | 854 | 15,890 |
| Grassland | 12,494 | 369 | 755,651 | 27,166 | 20,636 | 10,642 |
| Forest | 32,969 | 31,126 | 97,289 | 152,999 | 79,731 | 48,281 |
| Agricultural Land | 4883 | 429 | 15,399 | 146 | 352,753 | 47,346 |
| Built-up Area | 232 | 108 | 149 | 104 | 141 | 104,727 |

Changes to LULC categories from 2020 to 2040 are shown in Table 8. It is predicted that 2.27% of the forest cover will change into a waterbody, 0.78% into sand or dry river bed, 4.63% into grassland, 5.4% into agricultural land, and 2.36% into a built-up area, with the remaining 84.5% remaining unchanged. Similarly to this, there is a 74.08% chance that agricultural land would continue in its current classification and a 2.16, 8.38, and 13.84% chance that it would be shifted into waterbodies, forests, and built-up areas, respectively. Moreover, the chances of conversion of the area under sand or dry river bed into waterbody and grassland are 19.59 and 28.04%, respectively. The chances of the built-up area remaining unchanged are expected to stay at 19.34% during this predicted period.

The research also reveals that more than 127,744 pixels are anticipated to change into other categorized classes from the forest cover, whereas roughly 699,213 pixels of the forest cover are anticipated to remain stable. Grassland and agricultural land will be the primary changes to the forest class. Additionally, 75,816 pixels are anticipated to remain

unchanged in the grassland class. Almost 58,272 pixels of the agricultural land class would be converted into the built-up area class. The waterbody class is anticipated to mostly shift to sand or dry river beds (40,170 pixels), while 45,272 pixels are anticipated to stay the same. Furthermore, it is anticipated that 33,863 pixels will be converted to the grassland class, and 49,474 pixels from the sand class will become pixels representing waterbodies. The built-up class has maximum stability; 52,754 pixels are predicted to remain unchanged. It is anticipated that approximately 113,180 pixels from agricultural land and 65,859 pixels from forest would be significantly transformed into a built-up class (Table 9).

**Table 8.** Probability of LULC changes in 2040 by percentage.

| 2020/2040 | Waterbody | Sand/Dry River Beds | Forest | Grassland | Agricultural Land | Built-Up Area |
|---|---|---|---|---|---|---|
| Waterbody | 21.69 | 23.7 | 9.41 | 29.84 | 7.15 | 8.21 |
| Sand/Dry river beds | 19.59 | 23.04 | 11.00 | 28.04 | 8.43 | 9.89 |
| Forest | 2.27 | 0.78 | 84.55 | 4.63 | 5.4 | 2.36 |
| Grassland | 8.05 | 7.65 | 30.50 | 17.14 | 25.58 | 11.08 |
| Agricultural Land | 2.16 | 0.85 | 8.38 | 0.7 | 74.08 | 13.84 |
| Built-up Area | 00.08 | 0.01 | 0.02 | 0.03 | 0.01 | 19.34 |

**Table 9.** Expected transition in 2040 by number of pixels.

| 2020/2040 | Waterbody | Sand/Dry River Beds | Forest | Grassland | Agricultural Land | Built-Up Area |
|---|---|---|---|---|---|---|
| Waterbody | 45,272 | 49,474 | 19,645 | 62,296 | 14,932 | 17,144 |
| Sand/Dry river beds | 40,170 | 47,237 | 22,560 | 57,487 | 17,278 | 20,280 |
| Forest | 18,788 | 6483 | 699,213 | 38,279 | 44,637 | 19,557 |
| Grassland | 35,617 | 33,863 | 134,929 | 75,816 | 113,161 | 49,008 |
| Agricultural Land | 9079 | 3571 | 35,274 | 2927 | 311,832 | 58,272 |
| Built-up Area | 296 | 190 | 159 | 150 | 180 | 52,754 |

From 2020 to 2050, it is expected that forest cover will have a 78.93% chance of staying the same and a 2.81, 1.42, 5.33, 8.27, and 3.25% chance of changing into a waterbody, sand, grassland, agricultural land, or a built-up area, respectively. To the same extent, 67.04% of agricultural land is expected to stay the same, while the remaining percentages range from 2.85% to 12.96% to 1.63% to 13.45% for transformation into a body of water, sand, forest, grassland, or built-up area, respectively. There is a 1.61% chance that the body of water will remain a body of water. With respective probabilities of 17.63, 17.65, 24.19, 14.51, and 9.92%, it could be transformed into sand, forest, grassland, agricultural land, or a built-up area. The chances of the grassland becoming a waterbody or sand are 7.33% and 6.75%, respectively. Furthermore, this work assigns a 13.45% likelihood that built-up land will not change (Table 10).

**Table 10.** Probability of LULC changes in 2050 by percentage.

| 2020/2050 | Waterbody | Sand/Dry River Beds | Forest | Grassland | Agricultural Land | Built-Up Area |
|---|---|---|---|---|---|---|
| Waterbody | 16.1 | 17.63 | 17.65 | 24.19 | 14.51 | 9.92 |
| Sand/Dry river beds | 15.22 | 16.7 | 19.01 | 22.82 | 15.85 | 10.41 |
| Forest | 2.81 | 1.42 | 78.93 | 5.33 | 8.27 | 3.25 |
| Grassland | 7.33 | 6.85 | 34.76 | 11.75 | 29.04 | 10.27 |
| Agricultural Land | 2.85 | 1.55 | 12.96 | 1.63 | 67.04 | 13.97 |
| Built-up Area | 0.06 | 0.09 | 0.01 | 0.03 | 0.02 | 13.45 |

The analysis also reveals that more than 174,236 pixels are anticipated to change from forest classes to the other class, whereas 652,722 pixels of the forest cover are anticipated to remain stable. Grassland and built-up areas would be the primary changes to the forest class. Approximately 138,744 pixels are anticipated to transition from other classes into

cultivated land, while approximately 282,212 pixels in agricultural land are anticipated to remain stable. The grassland (6860 pixels) and the built-up areas (58,813 pixels) would be significantly transformed from agricultural land. It demonstrates that 175,156 pixels are anticipated to move from the waterbody category to other classes, while 33,607 pixels of the waterbody class are expected to remain unaltered. Around 34,229 pixels are projected to remain in the sand class, and more than 170,784 pixels are anticipated to change from the sand area to the other class. It is anticipated that 36,676 pixels for the built-up area class will remain unchanged (Table 11).

**Table 11.** Expected transition in 2050 by number of pixels.

| 2020/2050 | Waterbody | Sand/Dry River Beds | Forest | Grassland | Agricultural Land | Built-Up Area |
|---|---|---|---|---|---|---|
| Waterbody | 33,607 | 36,808 | 36,838 | 50,500 | 30,297 | 20,713 |
| Sand/Dry river beds | 31,208 | 34,229 | 38,964 | 46,777 | 32,491 | 21,344 |
| Forest | 23,230 | 11,704 | 652,722 | 44,079 | 68,365 | 26,858 |
| Grassland | 32,434 | 30,311 | 153,792 | 51,977 | 128,467 | 45,412 |
| Agricultural Land | 11,994 | 6514 | 54,563 | 6860 | 282,212 | 58,813 |
| Built-up Area | 262 | 185 | 114 | 103 | 170 | 36,676 |

### 3.3.2. Spatio-Temporal Analysis of the Predicted LULC

This study used LULC maps to generate the transition probability matrix, which was then used to generate LULC prediction maps using the CA-Markov model (Figure 8). The study showed that by the end of 2030, 2040, and 2050, the area covered with waterbodies would remain at 6.39, 5.44, and 4.94%, and the area covered with sand or dry river beds remain at 7.82, 6.74, and 9.57%. The forecasting of forest and grassland showed they would continue to cover a significant amount of area, which is 29.49 and 30.51 in 2030, 28.49 and 28.12 in 2040, and 25.08 and 30.27 in 2050. Agricultural land more or less remains the same within the predicted years, which is 11.81, 14.68, and 11.53 in 2030, 2040, and 2050. Despite having an eco-sensitive zone, the area shows a significant increase in the built-up area class of 13.95, 16.51, and 18.58% in these predicted years. The most stable land-use class is among the LULC classes' built-up area (Table 12). Between the years 2030 and 2050, it was predicted that there would be a net loss of water, sand, and forest cover, while an increase in grassland, agricultural land, and built-up areas would occur. Analysis of LULC projections for the next 30 years shows that there will be significant changes to the overall area and a large portion of land will be developed into built-up areas. The Core KNP region is protected by the Wildlife Protection Act of 1972, so the area's ideal mix of forest and grassland shall remain preserved.

**Table 12.** Area of the predicted LULC classes of the study area.

| LULC Classes | 2020 (Actual) | | 2030 (Predicted) | | 2040 (Predicted) | | 2050 (Predicted) | | Change from 2020–2050 | |
|---|---|---|---|---|---|---|---|---|---|---|
| | Area (km$^2$) | Area (%) | Area (km$^2$) | Area (%) | Area (km$^2$) | Area (%) | Area (km$^2$) | Area (%) | Area (km$^2$) | Area (%) |
| Waterbody | 187.50 | 8.61 | 139.14 | 6.39 | 118.61 | 5.45 | 107.62 | 4.94 | 79.88 | 3.67 |
| Sand/Dry river beds | 293.66 | 13.48 | 170.47 | 7.83 | 146.78 | 6.74 | 208.57 | 9.58 | 85.09 | 3.90 |
| Forest | 701.34 | 32.19 | 642.15 | 29.49 | 620.45 | 28.50 | 546.27 | 25.09 | 155.07 | 7.10 |
| Grassland | 496.26 | 22.78 | 664.38 | 30.51 | 612.26 | 28.12 | 659.13 | 30.27 | −162.87 | −7.49 |
| Agricultural Land | 143.12 | 6.57 | 257.18 | 11.81 | 319.64 | 14.68 | 251.15 | 11.53 | −108.03 | −4.96 |
| Built-up Area | 356.97 | 16.38 | 303.92 | 13.95 | 359.52 | 16.51 | 404.59 | 18.58 | −47.62 | −2.20 |

### 3.3.3. Validation and Evaluation of the Predicted LULC Map

The focus discrepancy between the actual LULC image and the simulated LULC image was caused by allocation errors rather than quantity issues. The values of Agreement

Chance, Agreement Quantity, Agreement Grid Cell, Disagreement Grid Cell, and Disagreement Quantity in Table 13 offer statistical information about the agreement between the simulated map and the reference map. To identify the model's simulated results, Disagreement Grid Cell and Disagreement Quantity elements are essential. In general, there is little difference between the two maps, and this is primarily because of quantity errors (0.0104) rather than allocation errors (0.0047). The agreement metrics reveal generally good agreement (98.6%) between the actual and simulated maps. Thus, the model's capacity to forecast LULC changes in location as opposed to quantity is stronger in the research area. This shows that the model has a good capability for simulating the states of future LULC and accurate location definitions.

**Table 13.** Validation analysis of two images (agreement and disagreement components values).

| Agreement/Disagreement | Value (%) |
|:---:|:---:|
| AC * | 14.59 |
| AQ | 07.66 |
| AS | 0.00 |
| AG | 76.55 |
| DG | 0.47 |
| DS | 0.00 |
| DQ | 01.04 |

* *AC* Agreement Chance, *AQ* Agreement Quantity, *AS* Agreement Strata, *AG* Agreement Grid Cell, *DG* Disagreement Grid Cell, *DS* Disagreement Strata, *DQ* Disagreement Quantity.

The predicted expected data (LULC) from the Markov model were compared with the actual LULC from 1990 to 2020. The computed value of $X^2$ is 4.24, which is significantly less than the significance of 11.070 on the critical area of 0.05 with five degrees of freedom (Table 14). With the acceptance of the hypothesis, it can be said that the actual transition probability of the 1990–2020 matrix is fitted with the anticipated transition probability created using the Markov method. The expected transition probability calculated using the Markov approach is similar to the actual transition probability of the matrix from 1990 to 2020. As a result, the CA-Markov model offered a superb fit for the projected LULC for the study area.

**Table 14.** Validation of the change prediction based on actual and predicted 2020 LULC image.

| LULC Classes | Area of the Predicted LULC 2020 (Km$^2$) | Area of the Actual LULC 2020 (Km$^2$) |
|:---:|:---:|:---:|
| Waterbody | 200.4 | 187.5 |
| Sand/Dry river beds | 280.8 | 293.66 |
| Forest | 710.23 | 701.34 |
| Grassland | 505.56 | 496.26 |
| Agricultural Land | 151.74 | 143.12 |
| Built-up Area | 330.12 | 356.97 |

Note : $X^2 = \sum \frac{(P-A)^2}{A} = 4.24$; df = 5 and $X^2_{0.05}$ (5) = 11.070. Data shown in the 2nd and 3rd columns as in P and A represent the absolute area in km$^2$ of individual LULC classes used for model validation.

### 3.4. Land Use and Land Cover Transformation and Temperature-Precipitation Changes

The Kaziranga Eco-Sensitive Zone was monitored by classifying land use and land cover consistently over time from 1990 to 2020. There could be a variety of factors that have ruled these shifts [33]. The LULC has changed significantly [13] due to both natural- and human-caused factors. The study area is situated in a low-lying area close to the Brahmaputra River, whose annual flooding is a major factor in the region, thus affecting the critical ecosystems of the region [34]. In times of extreme flooding, as much as 90% of the park's central area can be submerged [35]. During the monsoon season, the majority of the national park is underwater for about two to three months [36]. The grassland expands, and the wetland's water supply is replenished as a result of the massive flood's

nutrient inflow and vegetation succession [37–41]. Simultaneously, the annual flood causes significant siltation in the area, changing the wetland's overall outlook. Waterlogging has developed and occupied grassland areas in the national park as a result of the siltation of small channels. The grasslands are transformed into swamps, marshes, and waterlogged wetlands as a result of siltation. Land use and land cover changed in the region due to the shifting Brahmaputra River's bank line [14,42].

*3.5. Future Prediction of LULC in the Year 2030, 2040, and 2050*

To conduct a study on future land use land cover scenarios, in this research, the CA-Markov model was used to perform the LULC prediction scenarios for the years 2030, 2040, and 2050. Figures 7 and 8 display the predicted change in different LULC classes from 2030 to 2050. Creating a simulation of the area's future is crucial to its progress and long-term viability. It is essential to assess and enhance progress in the field regularly [43]. Additionally, this research will assist in forming environmental policies for upcoming projects using the simulation results. The total predicted class-wise area and percentage in this study are given in Table 12. The results demonstrate the gradual changes during the predicted study period. The built-up areas will continuously increase from 2030 to 2050. This steady growth will be due to the rising population and the increase in tourist flow to the region. Grassland also shows a significant increase in the predicted period from 2030 to 2050. The area with grassland will increase between 2030 and 2050.

Moreover, the area covered with agricultural land will increase from 2030 to 2050. The increasing trend in agriculture is due to the increasing population in the region. More population means more demand for food grains, which is associated with more regional agricultural expansion [44–47]. Forest surface will decrease between 2030 and 2050 because of the expansion of agricultural land and built-up surfaces. The simulated maps also show an alarming scenario for the waterbody class, which will continuously decrease from 2030 to 2050 because of its conversion into sand or dry river beds. This scenario is worse than the built-up area expansion in the study area [48]. The increasing area of sand or dry river beds would harm the natural ecosystems and surface water holding capacity [49–54].

## 4. Conclusions

The eco-sensitive zone surrounding Kaziranga has undergone numerous changes over the past 30 years due to both human and natural causes. Several LULC classes in the area exhibit a notable anomaly. Between 1990 and 2020, nearly 264.39 km$^2$ of agricultural land was lost. Most agricultural land has been transformed into urban areas. The built-up areas in the surrounding eco-sensitive zone are continuously expanding, despite all the rules and restrictions. This increasing trend of built-up areas is associated with development around the KNP.

In addition, the NH-37 passes through the heart of the Kaziranga Eco-Sensitive Zone, and it is along this highway that, due to development, the population has begun to increase rapidly. Because of this, less agricultural land and forest have been left. From 2010 to 2020, the study area was negatively impacted by the expansion of the built-up area in the southern part of the region, which is primarily bounded by the Karbi Anglong hills. The expansion of sand and dry river beds is a major problem, especially in the northern part of the study region. This has led to a decline in the size of the body of water over the past three decades. If the current trend continues, the built-up area, sand or dry river beds, and grassland are expected to increase, while waterbodies, forests, and agricultural land will decrease.

Human interference, and urban sprawl, in particular, is a major issue in the Kaziranga Eco-Sensitive Zone. From 1990 to 2020, there has been consistent growth in the built-up areas. The growth of human settlements obstructs animal migration routes, reduces wildlife habitat, and cuts off the eco-sensitive zone from the rest of the natural environment. Forests and grasslands have persisted in the Kaziranga Eco-Sensitive Zone throughout the time period under study, despite numerous threats and shifts in the environment. The initiatives

and policies of the state government have assisted in the restoration of lost patches of forest and grassland.

The forest cover and grassland cover of the Kaziranga Eco-Sensitive Zone were preserved throughout the study period despite numerous threats and changes. There has been some success in reestablishing forests and grasslands in areas where they had been lost, thanks to initiatives and policies implemented by state governments. Upon the order of the Guwahati High Court, Assam's state government began an eviction program in 2015, clearing 600 households from the area around Kaziranga National Park. The eco-sensitive zone around Kaziranga can expand and reclaim more forest thanks to this eviction. It is good news for the local ecosystem that some of the agriculture will be converted into grassland after the eviction process.

**Author Contributions:** Conceptualization, N.N., D.S., J.D., D.L. and P.K.; methodology, G.M., N.N., D.S., P.K. and M.F.; software, G.M., validation, G.M., S.K., N.N., D.S., J.D., D.L.; formal analysis, N.N., D.S., J.D., D.L., P.K. and S.K.S.; investigation, S.K., N.N., D.S., J.D., D.L. and M.F.; resources, D.S. and P.K.; data curation, N.N., D.S., J.D., D.L., K.C. and P.C.; writing—original draft preparation, N.N., D.S., J.D., D.L., K.C. and P.C.; writing—review and editing, G.M. and N.N.; visualization, G.M., S.K., N.N., D.S., J.D., D.L. and M.F.; supervision, P.K. and D.S.; project administration, D.S., J.D., D.L. and S.K.S.; funding acquisition, P.K. All authors have read and agreed to the published version of the manuscript.

**Funding:** This research received no external funding.

**Data Availability Statement:** The data will be available on relevant request from the corresponding author.

**Acknowledgments:** The author J.D. is grateful to the University Grants Commission, New Delhi for granting him the D.S. Kothari Post-Doctoral Fellowship (UGC-DSKPDF) [Enrolment Number: F.4-2/2006(BSR)/ES/20-21/0008]. The corresponding author G.M. is thankful to the Department of Science and Technology, Government of India, for providing a Fellowship under the Scheme for Young Scientists and Technology (SYST-SEED) [Grant no. SP/YO/2019/1362(G) and (C)]. The authors are grateful to all the anonymous reviewers whose valuable suggestions improved the quality of this manuscript.

**Conflicts of Interest:** The authors declare that this research work is original and have duly acknowledged all the sources of information used in the paper. There are no conflict of interest.

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
