# Peer review of "Land Use and Land Cover Change Monitoring and Prediction of a UNESCO World Heritage Site: Kaziranga Eco-Sensitive Zone Using Cellular Automata-Markov Model"

_land, doi:10.3390/land12010151_

Round 1
Reviewer 1 Report
Based on the historical land use laws from 1990 to 2020, this paper conducted a simulation experiment about the land use change before 2050. The research conclusions are helpful for the revision of urban planning and development strategies in the region. However, the following major issues need to be addressed before this paper is published:
Line 12, a yellow label after the centre should be deleted[1].
Line 24, What is the KNP?
Line 28, Remote should be capitalized the first letter.
Line 42-44, the accuracy result was not introduced clearly, it is suggested to supplement some quantitative results.
In the review section, Although the author has carried out a detailed review of related research, some newest relevant literature is still missing. It is recommended that the author add some latest literature, to well prove the advance of this paper and compare the differences with existing research. For instance, the following list, but not limited to these.
https://doi.org/10.3390/land11091598
https://doi.org/10.1016/j.scs.2022.104055
https://doi.org/10.3390/ijerph191912198
Line 109, 2214 sq. km. ->km2
Line 180, the formula lacks the necessary variable explanation, the same problem also occurs in the formulas 3 and 4.
Line 220, This line is a bit confusing.
Figure 2, the procedure LULC maps 2010 and 2020, an arrow may be missing here. Moreover, the procedure “integrated”, integrated what? this position looks like a procedure, so an adjective may not present the true meaning.
Line 297, 2000and2010, spaces may be missing.
L302-316, The content of this part is too redundant and duplicates a lot of the content in Table 2. It is suggested that the author can refine this part of the content and summarize the distinctive parts.
It seems that this paper is missing a discussion section, in which you should clarify some issues, such as, what is special about land use simulation in the Kaziranga Eco-sensitive zone? What are the shortcomings or characteristics of the current research compared with similar research? But it is not limited to these.
There are too many abbreviations in the text, it is recommended to add a list of abbreviations in the appendix at the end of the text.
Author Response
General Comment: Based on the historical land use laws from 1990 to 2020, this paper conducted a simulation experiment about the land use change before 2050. The research conclusions are helpful for the revision of urban planning and development strategies in the region. However, the following major issues need to be addressed before this paper is published:
Response: We thank the worthy reviewers for critically analyzing our manuscript and suggesting valuable suggestions that have improved the quality of our manuscript. We have carefully incorporated all the suggestions you provided in your review, and we thank you once again. Please find below point-by-point responses to each comment provided in the earlier version of our manuscript.
Comment 1: Line 12, a yellow label after the centre should be deleted.
Response 1: We have removed the yellow label from line 12.
Comment 2: Line 24, What is the KNP?
Response: We have included the full form of KNP (Kaziranga National Park) in revised manuscript.
Comment 3: Line 28, Remote should be capitalized the first letter.
Response 3: We have revised it.
Comment 4: Line 42-44, the accuracy result was not introduced clearly, it is suggested to supplement some quantitative results.
Response 4: We have added the accurate estimation of the LULC changes in the abstract. The revisions are shown as track changes. Thanks again for helping us to improve our manuscript.
Comment 5: In the review section, Although the author has carried out a detailed review of related research, some newest relevant literature is still missing. It is recommended that the author add some latest literature, to well prove the advance of this paper and compare the differences with existing research. For instance, the following list, but not limited to these.
https://doi.org/10.3390/land11091598
https://doi.org/10.1016/j.scs.2022.104055
https://doi.org/10.3390/ijerph191912198
Response 5: We have added the latest relevant literature of this study, as suggested. The revisions are highlighted in the revised manuscript.
Comment 6: Line 109, 2214 sq. km. -> km2
Response 6: We have corrected the line. Thanks.
Comment 7: Line 180, the formula lacks the necessary variable explanation, the same problem also occurs in the formulas 3 and 4.
Response 7: We have added the necessary variable explanation in line 180 and also added the variable explanation of formulas 3 and 4. We highlighted the revised line in the revised manuscript.
Comment 8: Line 220, This line is a bit confusing.
Response 8: We have revised the line for better understanding. Thank you for the suggestion.
Comment 9: Figure 2, the procedure LULC maps 2010 and 2020. An arrow may be missing here. Moreover, the procedure "integrated", integrated what? this position looks like a procedure, so an adjective may not present the true meaning.
Response 9: We have added the arrow in figure 2. The procedure 'integrated' is not fitted for this methodological chart, and we have corrected it. Indeed, it was an error. We are grateful and obliged for pointing out this mistake. Thanks.
Comment 10: Line 297, 2000and2010, spaces may be missing.
Response 10: We have added the space in this line. Thanks.
Comment 11: L302-316, The content of this part is too redundant and duplicates a lot of the content in Table 2. It is suggested that the author can refine this part of the content and summarize the distinctive parts.
Response 11: We have refined this part and summarized it according to your suggestion. The revised part is highlighted in the revised manuscript.
Comment 12: It seems that this paper is missing a discussion section, in which you should clarify some issues, such as, what is special about land use simulation in the Kaziranga Eco-sensitive zone? What are the shortcomings or characteristics of the current research compared with similar research? But it is not limited to these.
Response 12: We have included the discussion and also refined the queries suggested by you. Thank you for your valuable suggestion.
Comment 13: There are too many abbreviations in the text. It is recommended to add a list of abbreviations in the appendix at the end of the text.
Response 13: We have removed the abbreviations in the text and replaced them with the full form. Due to the reductions in the number of abbreviations from the main text, we didn't considered making an abbreviation appendix necessary. We again thank the worthy reviewers for all the comments, which have tremendously helped improve our manuscript.
Reviewer 2 Report
Dear authors,
I enjoyed reading your paper. It is timely work and dealing with the more recent climate change challenging, especially the impact of climate change on the natural heritage.
The paper is generally well-written and presented; however, I have some suggestions, such as:
1- You have shown us only three maps of changes during the time (1990-2000), (2000-2010) and (2010-2020). I think ten years period interval is too long. So, please make the interval sorter. I would suggest three years or at least five years intervals.
2- It would also be better to present two maps on a page rather than putting all the maps on one page so that readers can see the changes over time.
3- Is it possible to go before 1990? For example, you can use CORONA satellite images (1960s- 1970s) as it will improve your findings.
4- I know it might be difficult for you, but is it possible to carry out some ground truthing? Can you show us how things been changed over time?
5- You can also add charts showing the change in temperature or rainfall from (1990-2020) so readers can see the impact of climate change on the area.
All the best,
Author Response
General Comment: Dear authors, I enjoyed reading your paper. It is timely work and dealing with the more recent climate change challenging, especially the impact of climate change on the natural heritage. The paper is generally well-written and presented; however, I have some suggestions, such as:
Response: We thank the worthy reviewers for critically analyzing our manuscript and suggesting valuable suggestions that improved the quality of our manuscript. We have carefully incorporated all the suggestions you provided in your review, and we thank you again. Please find below point-by-point responses to each comment provided in the earlier version of our manuscript.
Comment 1: You have shown us only three maps of changes during the time (1990-2000), (2000-2010) and (2010-2020). I think ten years period interval is too long. So, please make the interval sorter. I would suggest three years or at least five years intervals.
Response 1: In this study, we have analyzed the decadal change of LULC for 30 years. The study region of the present work is protected by the Wild Life Protection Act, 1972, and has experienced very negligible changes. We have tried to analyze the LULC of the study area with a five-year gap before selecting ten years. However, because no significant changes were identified within the 5-year interval, we chose to analyze the data over a 10-year period. We appreciate your suggestion and thank you for your critical review.
Comment 2: It would also be better to present two maps on a page rather than putting all the maps on one page so that readers can see the changes over time.
Response 2: We have added the two maps on a single page according to your suggestion. Thank you for your comment.
Comment 3: Is it possible to go before 1990? For example, you can use CORONA satellite images (1960s-1970s) as it will improve your findings.
Response 3: We analyzed the different Landsat imageries (Landsat MSS) and CORONA satellite images to monitor the changes before 1990. But due to very low resolution and cloud coverage (CORONA), the area is very difficult to monitor before 1990. We really appreciate your valuable suggestion and thank you for your critical analysis of our paper.
Comment 4: I know it might be difficult for you, but is it possible to carry out some ground truthing? Can you show us how things been changed over time?
Response 4: We appreciate your suggestion regarding the ground truthing of our study. We have used an accuracy assessment to measure the validity of the results, showing reliable results for our study. Due to constraints in accessibility, we were unable to do the ground truthing as you suggested. I hope you will consider this research limitation.
Comment 5: You can also add charts showing the change in temperature or rainfall from (1990-2020) so readers can see the impact of climate change on the area.
Response 5: We have added rainfall and temperature variation charts from 1990-2020. This was highlighted in the revised manuscript. We again thank the worthy reviewers for all the comments, which have tremendously helped in improving our manuscript.
Reviewer 3 Report
The manuscript is of very limited scientific interest. The scope of the research is too narrow for attracting an international audience. Results are not placed in a broad context.
The manuscript does not meet the minimal requirements for a scientific publication.
The Introduction is ineffective in presenting the aims of the research, being only a confused summary of the techniques used in the study. The methods are not convincing. For example, use of chi-square tests applied to proportions is obviously wrong, since chi-square tests can be applied only to absolute frequencies.
Results presented in the text are a duplication of those reported in the tables (which, in turn, would be better presented as graphs). Discussion is mixed with results, but virtually absent. Conclusions are a summary of results with no true discussion.
The manuscript is very poorly presented. It is full of typos and references are not correctly indicated in the text (sometimes as authors and years instead of numbers).
The English is extremely poor.
Author Response
Comment 1: The manuscript is of very limited scientific interest. The scope of the research is too narrow for attracting an international audience. Results are not placed in a broad context.
Response 1: From the perspective of preserving biodiversity, the research region's size is crucial. In order to better comprehend the situation around the national park, the paper focuses on changes in land use and land cover scenarios during the past 30 years. Our paper also included a simulation of the scenario up to 2050, which we believe will assist the local authorities in making appropriate decisions for land management. We appreciate your thoughtful review of our paper. Thank you.
Comment 2: The manuscript does not meet the minimal requirements for a scientific publication.
Response 2: Your suggestions are valued, and the manuscript has been improved a lot after revisions. Our paper presents several significant findings in its topic area that have been overlooked by previous research. Additionally, we found that there is a dearth of literature on future scenario analysis in the research area. I'm thankful for your valuable suggestion.
Comment 3: The Introduction is ineffective in presenting the aims of the research, being only a confused summary of the techniques used in the study. The methods are not convincing. For example, the use of chi-square tests applied to proportions is obviously wrong since chi-square tests can be applied only to absolute frequencies.
Response 3: In order for the reader to understand the aims and significance of the study, we have improved the introduction section and included key information. We followed the tested methodology found in many research articles, which is what we believe is best suitable for this kind of research. We changed the chi-square test section, among other corrections, to the problems you pointed out. It was an error, no doubt. In the updated version, we have revised and modified that section. We appreciate your thoughtful feedback that has assisted us in the making our paper better in quality and content. We are grateful for your suggestions.
Comment 4: Results presented in the text are a duplication of those reported in the tables (which, in turn, would be better presented as graphs). Discussion is mixed with results but virtually absent. Conclusions are a summary of results with no true discussion.
Response 4: We have refined the results part and included the necessary information for the discussion of our results. We corrected and modified the discussion and conclusion according to your suggestion. The revised sections are shown in track mode change in the revised manuscript. Thank you.
Comment 5: The manuscript is very poorly presented. It is full of typos and references are not correctly indicated in the text (sometimes as authors and years instead of numbers).
Response 5: We have corrected and refined the manuscript as per your suggestions, and we hope it will now present our work properly. We also removed all the typos and corrected the reference part, as you mentioned, and the corrected sections are shown in track mode in the revised manuscript. Thank you.
Comment 6: The English is extremely poor.
Response 6: We have refined the manuscript and also checked the proofread our manuscript through a native English speaker. We again thank the worthy reviewers for all the comments, which have tremendously helped improve our manuscript.
Round 2
Reviewer 1 Report
This paper has done a good job of revising the questions raised in the last round. I think the paper has reached the level of being published in the Land journal. I think this paper can be accepted now.
Author Response
Thank you very much accepting our article.
Reviewer 3 Report
The manuscript is only of local interest.
Manuscript organization is very poor and largely illogical.
Many sections are completely superfluous or misleading.
There are obvious statistical mistakes and serious incongruences.
Numerical results presented in Tables are duplicated in the text.
The English is still very poor; for example. the authors use expressions “would be increased” instead of “will increase” for projections. The style is very convoluted, very far from scientific standards.
In the following I report detailed comments and corrections:
Lines 24-25: delete “ The region's total area is 2177 km2 which is covered with various flora and fauna and large grassland fields.”
25: In 1985, Kaziranga -> In 1985, the Kaziranga
26: delete “ declared Kaziranga National Park (KNP) a World Heritage Site”
29-35: Change as follows “The current study attempted to analyze the land use land cover (LULC) change in the Kaziranga Eco-Sensitive Zone using remote sensing data with CA-Markov models”
35-38: delete this part
41: The result shows that the area under -> The area covered with
42: lower case for sand
46-51: Delete this part
51-52: Change as follows: The outcome of this study are expected to be useful for the long-term management of Kaziranga eco-sensitive zone
57-61: Delete this part
63-76: Delete this part
77: delete “complicated”
78: as an effective, dependable, and affordable means for conservation and management-> as effective tool for conservation and management
79-87: Delete this part
87-90: rephrase as follows: Cellular Automata-Markov (CA-Markov) models are widely used to predict LULC since they combine the principles of cellular automata (CA) with Markov chains [9] and their rigorous structure is particularly effective in considering the natural complexity of the landscape [11].
94-99: rephrase as follows: In this study we explored past and future LULC changes in the Kaziranga Eco-sensitive zone, an area which includes the Kaziranga National Park and a 10 km buffer area using CA-Markov models.
96-103: Delete this part
104-113: reshape as follows: Because of constant human influence, this area is slowly losing its natural characteristics. In particular, because of growing tourism activity, the southern part of the region now sees an increase in the surface occupied built-up areas and roads, which represent a serious disturbance to the free movements of the animals [13]. We hope that the findings of this study will help the relevant authorities and policymakers to practice more environmentally responsible land and natural hazard management.
113-115: Delete this part
118-119: The study area, Kaziranga eco-sensitive zone, lies within the latitudinal extent of 26.27 N to 26.51 N and the longitudinal extent of 92.54 E to 93.42 E in -> The Kaziranga eco-sensitive zone lies at the intersection of the
121: delete “which … India”
128: lower case for sand
129: division?
129: interface?
130-131: from 7 to 38 °C
134: lower case for sand
143: delete “primarily”
143: four types: four main types
146: the tall grasses and reeds of the higher ground make -> tall grasses and reeds make
150-151: Delete this part
153-155: change as follows: Landsat TM 1990, TM 2000, ETM+ 2010, and Landsat OLI 2020 were used (Table 1). The freely…
164: applied the techniques of -> included
173-4: Delete this part
175: lower case for S
181-2: Delete this part
182-8: Change as follows: Temporal changes in LULC were monitored and analyzed with a post-classification change detection method by employing a change matrix (raster polygon). The change matrix depicts the LULC change in each period starting from 1990 to 2020. LULC changes for the years 2030, 2040, and 2050 were also predicted.
194: lower case for where
194: correct C-i
201: lower case for where
204-208: delete. Start with “The accuracy”
213: delete “using this matrix table”
214-6: Change as follows: The overall accuracy is calculated as the total number of correctly classified pixels divided by the total number of pixels in the error matrix, whereas the Kappa coefficient defines the degree of agreement between the classified map and reference data [22], [23]. These two measures were computed using the following formulas:
220: lower case for where
224: lower case for where
228-234: Change as follows: We used CA-Markov models to simulate the most likely future scenarios of LULC changes [23], [24]. IDRISI Terrset software was applied to the transition suitability image [DEFINE!] and LULC maps from 1990, 2000, 2010, and 2020 [25], [26].
250: Change as follows: Expected shifts in LULC for the future are calculated as:
264: The low -> A low
264: the high -> A high
268-70: Delete this part
275: comprehend?
277: insights
Results and discussion: Despite my previous advice, the authors still present the same data in the text and in the tables. I show here how to avoid it, but I cannot rewrite the text entirely!
293: the forest cover began to shrink, reaching 28.73 percent in 2000 and 25.59 percent in 2010, correspondingly. In 2020, tree cover increased, reaching 32.19 percent of the area under study (Table 2) -> the forest cover began to shrink, but in 2020 increased (Table 2)
301-304: Grassland expansion occurred concurrently with forest expansion, particularly between 1990 and 2010. Grassland covered a larger percentage of the landscape between 1990 and 2010 (25.32 and 36.9 percent, respectively), but by 2020, that percentage had dropped to 22.78 percent -> Grassland expansion occurred concurrently with forest expansion, particularly between 1990 and 2010. Grassland covered a larger percentage of the landscape between 1990 and 2010, but by 2020, that percentage had dropped.
308: Delete “According to the results”
310-311: has been almost doubled from 1990to 2020.
316: Many LULC shifts occurred in the study area during the 1990s, the 2000s, and the 2010s. Alterations by year, from 1990 to 2020, are displayed in Table 3. -> Many LULC shifts occurred in the study area during the 1990s, the 2000s, and the 2010s (Table 3).
316-332: Change as follows: There was an increase in the area with sand and dry river beds, grassland, and built-up land between 1990 and 2010. The area covered with water bodies, forests, and agricultural land was reduced. In the period 2000 to 2010, the area occupied by sand/dry river beds, grassland, and built-up area expanded, whereas areas under the waterbody, forest, and agriculture were shrunk. Between 2010 and 2020, the area with waterbody, sand/dry river beds, forests, and built-up surfaces increased, while the areas covered with grassland and agricultural land decreased. Between 1990 and 2020, the amount of land covered with built-up surfaces, forest and sand/dry river beds increased, while the areas covered with water, grassland, and agricultural land shrank.
333: Table 3. Period-wise Transformation of LULC class of Kaziranga eco-sensitive zone -> Table 3. Period-wise transformations of LULC class in the Kaziranga eco-sensitive zone
338-347: delete numerical values of surfaces: they are given in the table!
338-347: replace ‘under’ with ‘covered with’
348: Table 4. Period-wise Transformation of LULC class of Kaziranga eco-sensitive zone -> Table 4. Period-wise transformations of LULC class of the Kaziranga eco-sensitive zone
Table 4: Agricultural Land-Grassland row, last column: 4 -> 4.00
355-8: Delete this part
358-360: Change as follows: In all LULC categorized maps, both kappa accuracy overall accuracy retained an accuracy of 85 percent and above, which is reliable for further analysis [20,31].
Table 5: Why are different numbers of decimals used?
370: the notifications -> the changes
372 (t2) is repeated two times, please check
All the section from line 376 to 477 suffers from this duplication of results between text and tables. Present numerical values only in the tables and adopt a more concise style for text as I have illustrated for lines 316-332 as an example.
Tables 6, 8, 10. These values are in contrast with those in the text! For example, 0.349 in table 6 is 3.49 in the text!!! I think that in the table you DO NOT HAVE PERCENTAGES, but proportions. If so, replace numerical values in the tables with percentages.
487: Capital for validation
488-90: Delete this part
Table 13: Delete the Value column, only percentages are needed. Use two decimals. Note that there is inconsistence in the first row: 0.1459 is not 14.39% but 14.59%
492: Delete: The result demonstrates that
494: issues. -> issues (Table 13).
501: The outcome demonstrates that -> Thus,
505: Delete this line. Start with: “The predicted…”
507: Delete “For these facts, the hypothesis is accepted”
Table 14: Delete the last line (Total) it is confounding and statistically wrong. Delete the last column ((P-A)2/A): it us unnecessary and confounding
522: at play -> that have a rule
527: was -> is
534: delete significantly
535-548: Delete this section (including Figure 9). The used statistics are inappropriate and results inconsistent.
549: delete (8.67%) and (projected)
550: delete (16.38 percent)
548-569: All this part should be moved to the Conclusions section
572: scenario->scenarios
574: delete “in the Kaziarga eco-sensitive zone”
578: are in Table -> are given in Table
578: delete “calculated”
580: grown from 303.92 km2 in 2030 to 359.52 km2 and then 404.59 km2 in 2050. -> grown from 2030 to 2050.
583: area under grassland would be increased to 664.38 km2 in 2030 and 659.13 km2 in 2050 -> area with grassland will increase between 2030 and 2050.
584-595: Change as follows: Moreover, the area covered with agricultural land will increase from 2030 to 2040 2050. The increasing trend in agriculture is due to the increasing population in the region. More population means more demand for food grains, which is associated with more regional agricultural expansion [44-47]. Forest surface will decrease between 2030 and 2050 because of the expansion of the agricultural land and built-up surfaces. The simulated maps also show an alarming scenario for the water body surfaces, which will continuously decrease from 2030 to 2050 because of its conversion into sand/dry river beds. This scenario is worse than the built-up area expansion in the study area [48]. The increasing area of sand/dry river beds would harm the natural ecosystems and the surface water holding capacity [49].
596-7: delete this sentence
599: Delete “Researchers have found that”. Start with “The eco-sensitive”
602: apex for 2 in km2
602: developed -> transformed
604-605: delete this sentence
606-8: Delete this sentence
609-610: In addition to that, NH-37, or the National Highway, is a critical transportation corridor in the area. The NH-37 passes -> In addition, the NH-37 highway passes
618-9: increase to 404.59, 208.57, and 659.13 km2 by 2050, while water bodies, forests, and agricultural land will decrease by 107.62, 546.27, and 251.15 km2 -> increase, while water bodies, forests, and agricultural land will decrease.
620-9: Delete all this part
62-632: Change as follows: We hope that our results may assist policymakers and land managers responsible for urban and rural settings by providing them with information useful to improve land administration in protected areas.
Author Response
Dear Reviewer,
We sincerely thank you for all of your efforts on our manuscript. It has improved a lot as a result of your valuable suggestions. We have incorporated each and every suggestion you had provided in the report. The point by point responses sheet is attached for your perusal.
We sincerely hope that you shall endorse our manuscript for publication this time.
Best regards
Gowhar Meraj
